# Affecting Factors and Correction Ratio in Genu Valgum or Varum Treated with Percutaneous Epiphysiodesis Using Transphyseal Screws

**DOI:** 10.3390/jcm9124093

**Published:** 2020-12-18

**Authors:** Si-Wook Lee, Kyung-Jae Lee, Chul-Hyun Cho, Hee-Uk Ye, Chang-Jin Yon, Hyeong-Uk Choi, Young-Hun Kim, Kwang-Soon Song

**Affiliations:** Department of Orthopedic Surgery, Keimyung University Dongsan Hospital, Keimyung University School of Medicine, 1035 Dalgubeol-daero, Dalseo-gu, Daegu 42601, Korea; oslee@dsmc.or.kr (K.-J.L.); oscho5362@dsmc.or.kr (C.-H.C.); yhu4032@gmail.com (H.-U.Y.); poweryon@nate.com (C.-J.Y.); neojy0525@dsmc.or.kr (H.-U.C.); 190314@dsmc.or.kr (Y.-H.K.); skspos@naver.com (K.-S.S.)

**Keywords:** genu valgum, genu varum, correction rates, percutaneous, epiphysiodesis, transphyseal screws, pediatric surgery

## Abstract

This study evaluated the correction rates of idiopathic genu valgum or varum after percutaneous epiphysiodesis using transphyseal screws (PETS) and analyzed the affecting factors. A total of 35 children without underlying diseases were enrolled containing 64 physes (44 distal femoral (DT), 20 proximal tibial (PT)). Anatomic tibiofemoral angle (aTFA) and the mechanical axis deviation (MAD) were taken from teleroentgenograms before PETS surgery and screw removal. The correction rates of the valgus and varus deformities for patients treated with PETS were 1.146°/month and 0.639°/month using aTFA while using MAD showed rates of 4.884%/month and 3.094%/month. After aTFA (*p* < 0.001) and MAD (*p* < 0.001) analyses, the correction rate of DF was significantly faster than that of PT. Under multivariable analysis, the aTFA correction rate was significantly faster in younger patients (*p* < 0.001), in males (*p* < 0.001), in patients with lower weights (*p* < 0.001), and in the group that was screwed at DF (*p* < 0.001). Meanwhile, the MAD correction rate was significantly faster in patients with lower heights (*p* = 0.003). PETS is an effective treatment method for valgus and varus deformities in growing children and clinical characters should be considered to estimate the correction rate.

## 1. Introduction

Angular deformities of the knee at the coronal plane are relatively common causes for pediatric orthopedic surgeon referrals. They cause cosmetic problems, gait disturbances, joint instabilities with pathologic crepitus, and progressive activity-related pains in growing children [1]. Malalignment of the knee produces pathologic stress on the unicompartmental joint space of the knee, increasing the risk of accelerated degenerative changes [2]. The current recommendation addresses lower limb angular deformities using mechanical axis assessment as early as possible to prevent consequences such as altered gait patterns, early articular cartilage degeneration, and soft tissue changes [3,4,5,6].

Asymmetrical suppression of the physes in growing children has been proven to correct coronal angular deformities of the knee [7,8,9,10,11,12]. Percutaneous epiphysiodesis using transphyseal screws (PETS) can be a good treatment option for correcting coronal angular deformities because it gradually corrects deformities and can be terminated by screw removal after completion [13]. Measuring the average rate of correction is important when predicting the time for screw removal, and it allows patients and surgeons to expect when treatment will end. Identifying the factors that affect the rate of correction is also important for predicting treatment outcomes.

This study aims to evaluate the rate of angular correction after PETS for idiopathic genu valgum or varum and analyze the factors that can affect the rate of correction.

## 2. Patients and Methods

After obtaining approval from the Keimyung University Dongsan Hospital Institutional Review Board (KMUDSH IRB), this study reviewed the electronic medical records and radiographs of patients who had undergone epiphysiodesis around the knee for angular deformity from April 2008 to April 2016. The inclusion criteria for this study were children with idiopathic coronal angular deformity with valgus or varus knees treated with PETS, hemiepiphysiodesis of the distal femur (DF) or the proximal tibia (PT) using only one screw on one knee, and adequate clinical and radiographic follow-up until skeletal maturity or for a minimum of 6 months after screw removal. The exclusion criteria included the following: children with concurrent hemiepiphysiodesis of the DF and the PT; inadequate preoperative or postoperative radiographs available for review; any non-idiopathic reasons of angular deformity such as a multiple epiphyseal dysplasia (MED), Morquio, metabolic, congenital, and posttraumatic; and any other bony procedures such as osteotomies. Patients who had concurrent hemiepiphysiodesis of the DF and PT were excluded because the study was designed to determine which of the femoral or tibial physes affected the rate of correction.

After the medical record review, the following patient information was obtained: sex, weight, height, and age at the time of hemiepiphysiodesis, physes treated, and the size of the screw used for PETS. The anteroposterior and lateral radiographs on the knee [knee anteroposterior (AP) and lateral (Lat.)] were taken after surgery to confirm the position of the screws (Figure 1 and Figure 2). The standing anteroposterior lower extremity computed teleroentgenogram (Philips medical systems, Hamburg, Germany) was taken on the first visit and every follow-up visit with the patella facing forward. All coronal plane imaging was performed with the patella pointed straight ahead and the distances between the subject and the X-ray tube were standardized.

Anatomical tibiofemoral angle (aTFA), mechanical axis deviation (MAD; Figure 3), anatomical lateral distal femoral angle (aLDFA) and anatomical medial proximal tibial angle (aMPTA) were measured in teleroentgenograms taken before PETS surgery and before screw removal. The aTFA is the angle formed by the intersection of the mechanical axes of the femur and the tibia, and the MAD is the ratio of the width of the tibial articular surface that the mechanical axis line crosses. These terms are defined as such to avoid errors of magnification [14]. A value greater than 50% indicated a varus alignment. In comparison, a value of less than 50% indicated a valgus alignment of the knee with lateralization of the mechanical axis.

Statistical Analysis. R language version 3.3.3 (R Foundation for Statistical Computing, Vienna, Austria) and the T&F program version 3.0 (YooJin BioSoft, Korea) were used for all statistical analyses, expressing the data as a mean ± standard deviation or median (1st–3rd interquartile) for continuous variables according to the normality.

This study performed an independent two-sample *t*-test, or the Mann–Whitney U test, to compare the mean difference of the continuous variables between the valgus and varus groups. For the categorical variables, a Chi-squared test and a paired sample *t*-test to test were used the difference between treatment effects such as aTFA and MAD before and after treatments. Linear regression analysis reveals the effect of risk factors on the correction ratio of aTFA and MAD. For analyzing the independent effect of multiple risk factors, the multivariable analysis was performed using a backward stepwise variable elimination procedure to maximize the Akaike information criterion. The initial input risk factors for multivariable analysis use *p* < 0.2 derived from the univariable analysis. Moreover, analysis of covariance (ANCOVA) was performed to test the difference of correction ratios between the valgus and varus groups and the difference of correction ratios between the group with the screw in the DF and the group with the screw in the PT by controlling confounding covariates.

## 3. Results

A total of 35 healthy children without any underlying diseases comprised the study population. The study included 64 physes (44 DF and 20 PT), and a total of 64 knees were treated with the PETS technique. Afterward, 44 knees with genu valgum were treated with medial hemiepiphysiodesis (40 DF and 4 PT), and 20 knees with genu varum were treated with lateral hemiepiphysiodesis (4 DF and 16 PT). The average age of patients at the time of surgery was 11.29 ± 1.44 years and female (12.27 ± 1.29 years) had been treated about 2 years earlier than male patients (10.73 ± 1.22 years). The mean follow-up after surgery was 28.3 ± 20.0 months (range 5–85 months). The mean weight was significantly higher in the valgus group than in the varus group (*p* < 0.001). Meanwhile, the mean “Tip ratio Lat” was smaller in the valgus group than in the varus group (*p* < 0.001), and this value was calculated as a median because it does not follow the normal distribution (Table 1).

The aTFA’s mean amount of correction was 5.20° in the valgus group and 5.77° in the varus group (Figure 4), while the MAD’s mean amount of correction was 0.21 in the valgus group and 0.30 in the varus group (Figure 5). Both indicators show that PETS surgeries were effective for the valgus and varus groups (Table 2).

The cannulated screws were removed after an average of 8.09 months (range 4–43 months) and after angular correction. The correction rate was confirmed by the changes of aTFA (°) and MAD (%, percentage) by dividing the amount of angular correction by the correction duration, in months, before the screws’ removal. Before controlling the covariate in the two-sample *t*-test, only the correction rate of aTFA was faster in the valgus group than in the varus group, and there was no statistically significant difference in the correction rate of MAD. However, after controlling demographic factors such as sex, height, and weight, and surgical factors such as “Tip ratio AP and Lat” and the sizes of the screws, the correction rates of both aTFA and MAD were significantly faster in the valgus group than in the varus group. The estimated correction rates of the valgus and varus deformities for patients treated with PETS surgery were 1.146°/month and 0.639°/month using aTFA, and 4.884%/month and 3.094%/month using MAD after ANCOVA application, respectively (Table 3).

This study attempted to identify the factors affecting correction rates after PETS surgery in patients with coronal angular deformities through a linear regression model. The correction rate of aTFA was significantly faster in younger patients (*p* = 0.026) and the group screwed at the DF (*p* < 0.001) under univariable analysis. After the univariable analysis, a backward stepwise variable elimination procedure was applied to the multivariable analysis. The correction rate of aTFA was significantly faster in younger patients (*p* < 0.001), in the male group (*p* < 0.001), in patients with lower weight (*p* < 0.001), and in the group screwed in the DF (*p* < 0.001) (Table 4).

The MAD correction rate was significantly faster in younger patients (*p* < 0.001), in patients with lower weights (*p* = 0.008), and patients with lower heights (*p* = 0.001) under univariable analysis. The only factor affecting the correction rate of MAD was the height (*p* = 0.003) in multivariable analysis (Table 5).

## 4. Discussion

In this study, the correction rates were significantly faster in the valgus group than in the varus group as a result of analyzing the measured values using aTFA (*p* < 0.001) and MAD (*p* = 0.007; Table 3). This study shows that the speed of correction for angular deformity is affected by age, sex, and weight when measured by aTFA, and by height when measured by MAD.

Although the study showed significant results on the observed correction rates, the study had several limitations. First, because this was a retrospective study, the patients’ ages during screw placement and the follow-up periods were variable, which might lead to biases when assessing the effect of a transphyseal screw. The cases were not compared with an untreated group of knees with genu valgum or varum, which can serve as a control group.

Second, although the linear regression model was used to reveal the effects of risk factors on the correction ratio, the sample size was relatively small for the multivariable statistical analysis. The rate of correction by dividing the valgus and varus groups was also analyzed (Table 6 and Table 7). There remains a possibility of bias, however, because of the small number of samples used for analysis.

Third, this study included some patients who had fewer follow-ups than others. However, the correction of angular deformities in these patients occurred over a shorter duration, and the data were analyzed using a linear regression model, excluding the independent variables that caused multicollinearity. Therefore, it is reasonable to include these patients in this study.

Several studies have reported the rate of correction of coronal angular deformity after asymmetrical physeal suppression. Most existing literature only distinguished the DF and PT to analyze correction rates. The correction rates of coronal angular deformities were 0.40 to 1.0°/month at the DF [4,11,15,16,17,18,19,20,21,22] and 0.36 to 0.82°/month at the PT [4,15,17,18,19,20,21,22,23,24]. However, the rate of correction could be affected by the patient’s age at the time of hemiepiphysiodesis, as well as their sex, weight, height, the surgical methods used (staples, transphyseal screws, or flexible plate), etiology, and the treatment of physis. Sung et al. [25] selected the linear mixed model and described that the correction rates of valgus deformities of younger children at the DF, PT, and distal tibia (DT) were estimated at 0.71°/month, 0.40°/month, and 0.48°/month, respectively. There was no significant difference seen in the correction rates of the valgus deformities between the staple group, screw group, and permanent group at the DF and DT. Given the retrospective nature of this study, limitations were present, such as a deficiency of repeated measurements or longitudinal data. Therefore, a linear regression model was selected to consider various factors, such as age, sex, weight, height, treatment of physis, and screw size. Future studies can perform progressive randomized controlled trials with the longitudinal data of patients with coronal angular deformities.

The generally accepted percentages of the longitudinal growth contribution of pairs of physes for each long bone in the lower extremities are 30%, 70%, 55%, and 45% for the proximal femoral, distal femoral, proximal tibial, and distal tibial growth plates, respectively [26,27,28,29,30,31]. Courvoisier et al. [23] described that stapling at the PT required a longer correction duration than that of the DF. Cho et al. [18] reported that the amount of angular correction at the DF was significantly larger than at the PT. Nonetheless, the rate of correction was not significantly different between the DF and the PT after hemiepiphyseal stapling in patients with multiple epiphyseal dysplasia. This study showed that the estimated correction rates of aTFA at the DF and PT were 1.195°/month and 0.600°/month after ANCOVA application, respectively. The correction rate of the DF was significantly faster than that of the PT as a result of analyzing the measured values using aTFA (*p* < 0.001) and MAD (*p* < 0.001; Table 8).

In conclusion, asymmetrical physeal suppression with percutaneous transphyseal screws for correcting the angular deformities of the knee is an effective method for treating both valgus and varus deformities in growing children. When treating coronal angular deformities, the age, height, and weight of the patient should be taken into consideration to predict the rate of correction accurately. Treatment should also consider that the rate of correction at the DF is faster than at the PT. The cosmetic aspect of these deformities cannot be disregarded, but the results of this study show the factors that determine the ideal time for young children to undergo deformity-correction procedures.

## Figures and Tables

**Figure 1 jcm-09-04093-f001:**
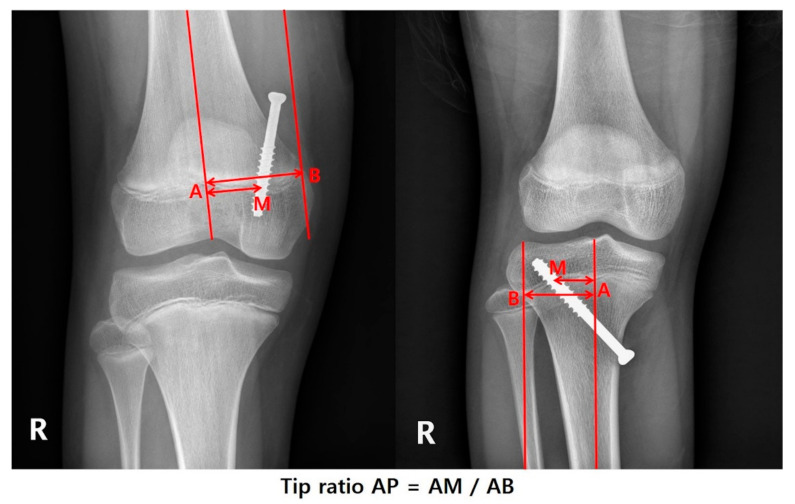
On the knee anteroposterior (AP) view, the screw position was determined as the ratio of AM (distance from center to screw) and AB (distance of half physis).

**Figure 2 jcm-09-04093-f002:**
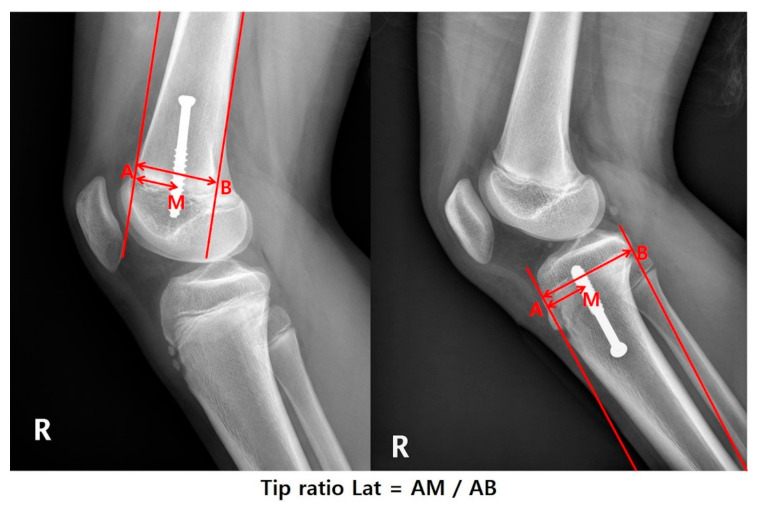
On the knee lateral view, the screw position was determined as the ratio of AM (distance from center to screw) and AB (distance of half physis). This ratio was named “Tip ratio Lat.”.

**Figure 3 jcm-09-04093-f003:**
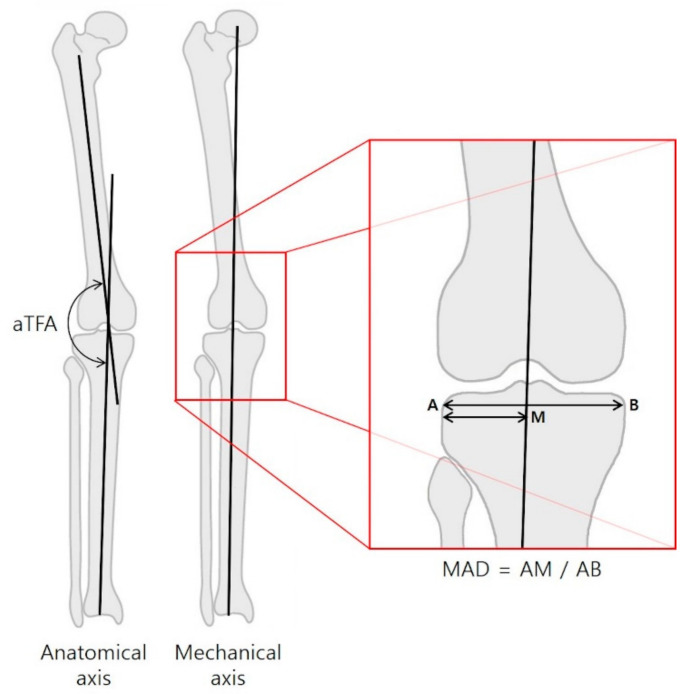
The anatomical tibiofemoral angle (aTFA) was measured on full-length standing radiographs. The mechanical axis deviation was expressed as a ratio of the distance of the axis from the lateral border of the tibia to the width of the tibial articular surface.

**Figure 4 jcm-09-04093-f004:**
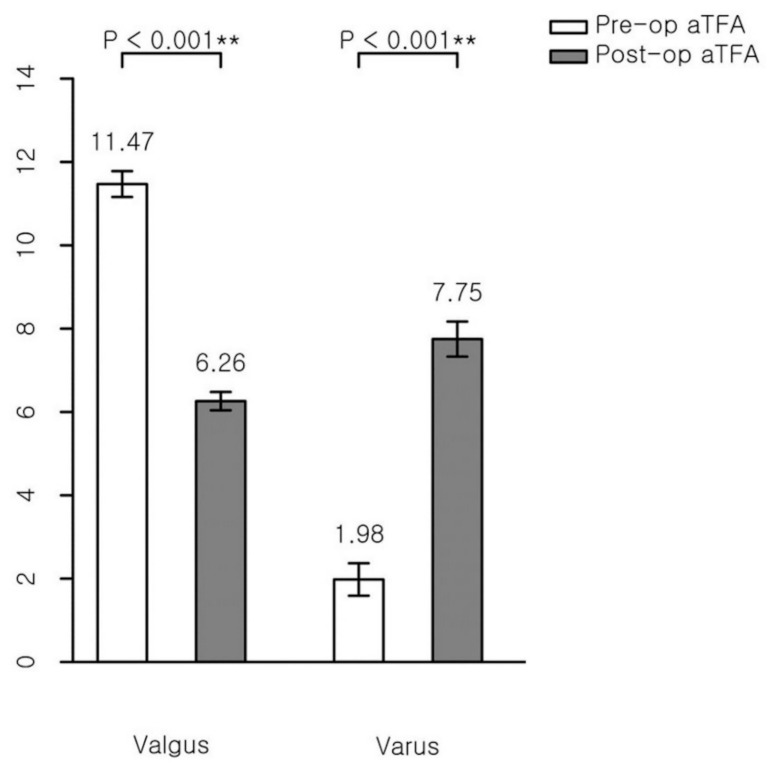
Mean amount of correction of anatomical tibiofemoral angle (aTFA). **, statistical value with significant difference (*p* < 0.001).

**Figure 5 jcm-09-04093-f005:**
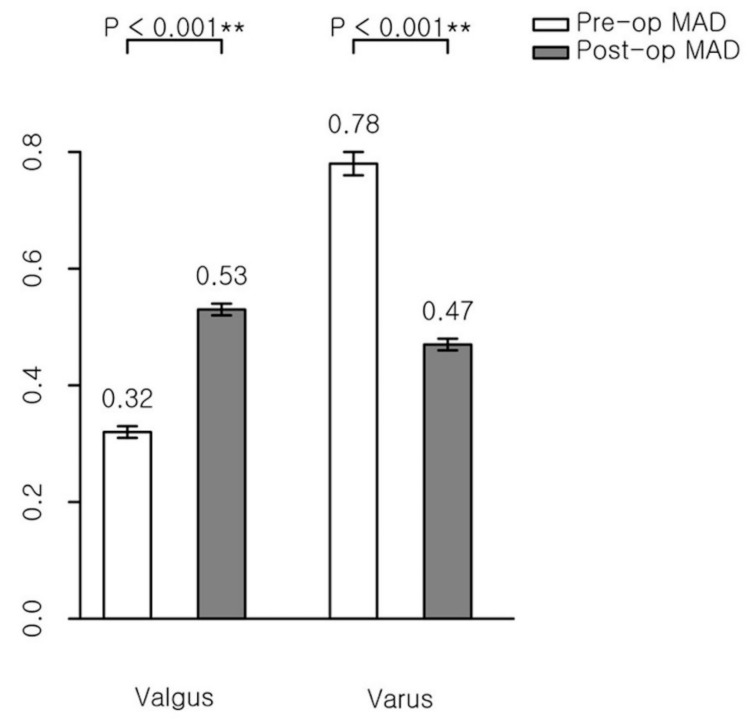
Mean amount of correction of mechanical axis deviation (MAD). **, statistical value with significant difference (*p* < 0.001).

**Table 1 jcm-09-04093-t001:** Patients’ demographics and summary of screw placements.

Variables	Total	Valgus	Varus	*p*-Value
Number of physes (%)	64 (100)	44 (68.75)	20 (31.25)	
Age at operation (months)	135.44 ± 7.33	133.52 ± 17.84	139.65 ± 15.77	0.192
Sex (male/female)	23/41	17/27	6/14	0.504
Weight (kg)	48.94 ± 13.69	52.47 ± 14.41	41.18 ± 7.74	<0.001
Height (cm)	151.25 (144.7–156.55)	151.9 (146.72–160.2)	149.55 (143.9–153)	0.126
BMI	21.5 ± 4.28	22.19 ± 4.12	19.99 ± 4.36	0.056
Limb (right/left)	33/31	23/21	10/10	0.866
Follow-up period (months)	19 (13–43)	19 (12–44)	19.5 (14.5–38)	0.994
Implantation duration (months)	6 (4–8)	5 (4–7)	9 (7–14)	<0.001
Tip ratio AP	0.64 ± 0.1	0.64 ± 0.09	0.63 ± 0.11	0.948
Tip ratio Lat	0.45 (0.39–0.5)	0.41 (0.37–0.48)	0.5 (0.47–0.54)	<0.001
Size of screw (7.0 mm/4.5 mm)	35/29	24/20	11/9	0.973

Continuous variables are expressed as mean ± standard deviation or median (1st–3rd interquartile) according to the normality of the variable. *p*-values for the continuous variables are computed using an independent two-sample *t*-test or Mann–Whitney U test according to the normality of the variable. *p*-values for the categorical variables are computed using a Chi-squared test.

**Table 2 jcm-09-04093-t002:** Comparison of mean difference between the preoperative and the postoperative indices.

Paired Variables	Pre-Op	Post-Op	Difference between Pre-Op and Post-Op	*p*-Value
aTFA (°)				
Valgus	11.47 ± 0.31	6.26 ± 0.22	−5.20 ± 0.32	<0.001
Varus	1.98 ± 0.39	7.75 ± 0.42	5.77 ± 0.64	<0.001
MAD (ratio)				
Valgus	0.32 ± 0.01	0.53 ± 0.01	0.21 ± 0.02	<0.001
Varus	0.78 ± 0.02	0.47 ± 0.01	−0.30 ± 0.03	<0.001
aLDFA(°)				
Valgus	78.78 ± 3.79	83.85 ± 3.64	−5.07 ± 4.15	<0.001
Varus	82.79 ± 2.67	81.06 ± 3.28	1.71 ± 4.78	0.127
aMPTA(°)				
Valgus	89.27 ± 2.76	88.74 ± 3.07	0.53 ± 2.88	0.227
Varus	85.26 ± 2.21	88.40 ± 3.03	−3.16 ± 4.19	0.003

aTFA, anatomical tibiofemoral angle; MAD, mechanical axis deviation; aLDFA, anatomical lateral distal femoral angle; aMPTA, anatomical medial proximal tibial angle.

**Table 3 jcm-09-04093-t003:** Comparison of correction rates between valgus and varus groups: Before and after adjustment by covariates.

Measurement of Correction Rate	Statistics	Valgus Group	Varus Group	*p*-Value
aTFA (°/month)	Two-sample *t*-test	1.03 ± 0.08	0.63 ± 0.08	0.003
	ANCOVA	1.146 ± 0.061	0.639 ± 0.095	<0.001
MAD (%/month)	Two Sample *t*-test	4.27 ± 0.39	3.31 ± 0.4	0.137
	ANCOVA	4.884 ± 0.324	3.094 ± 0.505	0.007

aTFA, anatomical tibiofemoral angle; MAD, mechanical axis deviation ANCOVA: analysis of covariance where sex, limb side, height, body mass ratio (BMI), Tip ratio AP, Tip ratio Lat, and screw size are used as covariates to be adjusted.

**Table 4 jcm-09-04093-t004:** Results of univariable and multivariable linear regression analysis using the correction rate (aTFA) as a response.

	Univariable Analysis			Multivariable Analysis		
Predictor (Response = Rate of Correction (aTFA))	Coef. (95% Cis)	St. Coef.	*p*-Value	Coef. (95% Cis)	St. Coef.	*p*-Value
* Age at operation (years)	−0.097 (−0.180 to −0.013)	−0.277	0.026	−0.135 (−0.202 to −0.067)	−0.387	<0.001
* Sex						
M:F	0.240 (−0.012 to 0.492)	0.231	0.066	0.669 (0.464 to 0.874)	0.644	<0.001
* Weight (kg)	−0.008 (−0.017 to 0.001)	−0.221	0.079	−0.019 (−0.025 to −0.012)	−0.507	<0.001
Height (cm)	−0.006 (−0.015 to 0.003)	−0.160	0.207			
BMI	−0.015 (−0.044 to 0.013)	−0.132	0.298			
* Screw Location						
PT:DF	−0.458 (−0.700 to −0.216)	−0.426	<0.001	−0.519 (−0.735 to −0.304)	−0.483	<0.001
Limb						
Rt:Lt	−0.084 (−0.331 to 0.164)	−0.084	0.510			
Tip ratio AP	−1.578 (−2.808 to −0.348)	−0.304	0.065			
Tip ratio Lat	−1.313 (−2.250 to −0.377)	−0.329	0.008			
Size of Screw (mm)						
4.5:7.0	0.101 (−0.147 to 0.349)	0.101	0.426			
Kind of Deformity						
Varus:Valgus	−0.398 (−0.647 to −0.149)	−0.370	0.003			

Coef, coefficient of linear regression; St. Coef, standardized coefficient of linear regression; Cis, confidence intervals. M, Male; F, Female; PT, proximal tibia; DF, distal femur; Rt, right; Lt, left. * Predictors were used as input variables for the multivariable analysis after backward stepwise variable elimination.

**Table 5 jcm-09-04093-t005:** Results of univariable and multivariable linear regression analysis using the correction rate (MAD) as a response.

	Univariable Analysis			Multivariable Analysis		
Predictor (Response = Rate of Correction (MAD))	Coef. (95% Cis)	St. Coef.	*p*-Value	Coef. (95% Cis)	St. Coef.	*p*-Value
* Age at operation (years)	−0.760 (−1.128 to −0.391)	−0.457	<0.001	−0.399 (−0.801 to 0.003)	−0.240	0.057
Sex						
M:F	0.588 (−0.639 to 1.815)	0.118	0.352			
Weight (kg)	−0.057 (−0.099 to −0.016)	−0.327	0.008			
* Height (cm)	−0.070 (−0.111 to −0.029)	−0.392	0.001	−0.069 (−0.113 to −0.025)	−0.385	0.003
BMI	−0.026 (−0.165 to 0.114)	−0.046	0.720			
* Screw Location						
PT:DF	−1.189 (−2.433 to 0.056)	−0.231	0.066	−1.055 (−2.373 to 0.264)	−0.205	0.122
Limb						
Rt:Lt	−0.015 (−1.201 to 1.172)	−0.003	0.981			
Tip ratio AP	−3.765 (−9.865 to 2.336)	−0.152	0.231			
Tip ratio Lat	−2.767 (−7.460 to 1.926)	−0.145	0.252			
Size of Screw (mm)						
4.5:7.0	1.060 (−0.101 to 2.222)	0.222	0.078			
Kind of Deformity						
Varus:Valgus	−0.967 (−2.223 to 0.290)	−0.188	0.137			

Coef, coefficient of linear regression; St. Coef, standardized coefficient of linear regression; Cis, confidence intervals M, Male; F, Female; PT, proximal tibia; DF, distal femur; Rt, right; Lt, left. * Predictors were used as input variables for the multivariable analysis after backward stepwise variable elimination.

**Table 6 jcm-09-04093-t006:** Results of univariable and multivariable linear regression analysis using the correction rate (aTFA) as a response in valgus and varus groups.

	Univariable Analysis				Multivariable Analysis		
Predictor (Response = Rate of Correction (aTFA))	Kind of Deformity	Coef. (95% Cis)	St. Coef.	*p*-Value	Coef. (95% Cis)	St. Coef.	*p*-Value
Age at operation (years)	Valgus	−0.101 (−0.200 to −0.002)	−0.294	0.052			
	Varus	−0.008 (−0.136 to 0.119)	−0.031	0.898			
** Sex							
M:F	Valgus	0.117 (−0.195 to 0.429)	0.113	0.466			
	Varus	0.439 (0.145 to 0.732)	0.568	0.009	0.333 (0.069 to 0.598)	0.432	0.024
Weight (kg)	Valgus	−0.018 (−0.027 to −0.008)	−0.495	<0.001			
	Varus	−0.001 (−0.023 to 0.021)	−0.018	0.941			
* Height (cm)	Valgus	−0.022 (−0.034 to −0.010)	−0.494	<0.001	−0.018 (−0.029 to −0.007)	−0.397	0.004
	Varus	0.002 (−0.008 to 0.013)	0.102	0.670			
BMI	Valgus	−0.034 (−0.070 to 0.003)	−0.270	0.076			
	Varus	−0.015 (−0.053 to 0.022)	−0.184	0.436			
Screw Location							
PT:DF	Valgus	−0.461 (−0.974 to 0.051)	−0.263	0.085			
	Varus	−0.231 (−0.626 to 0.163)	−0.261	0.266			
Limb							
Rt:Lt	Valgus	−0.088 (−0.393 to 0.216)	−0.087	0.572			
	Varus	−0.098 (−0.422 to 0.226)	−0.139	0.560			
* Tip ratio AP	Valgus	−2.519 (−4.045 to −0.993)	−0.447	0.053	−2.909 (−4.273 to −1.544)	−0.516	0.150
	Varus	−0.226 (−1.72 to 1.268)	−0.070	0.770			
** Tip ratio Lat	Valgus	−0.203 (−1.740 to 1.333)	−0.040	0.797			
	Varus	−1.543 (−2.507 to −0.58)	−0.595	0.006	−1.218 (−2.107 to −0.33)	−0.470	0.016
* Size of Screw (mm)							
4.5:7.0	Valgus	0.217 (−0.082 to 0.517)	0.214	0.163	0.223 (−0.046 to 0.493)	0.220	0.112
	Varus	−0.159 (−0.479 to 0.162)	−0.223	0.345			

Coef, coefficient of linear regression; St. Coef, standardized coefficient of linear regression; Cis, confidence intervals M, Male; F, Female; PT, proximal tibia; DF, distal femur; Rt, right; Lt, left * Predictors were used as input variables for the multivariable analysis after backward stepwise variable elimination in the valgus group. ** Predictors were used as input variables for the multivariable analysis after backward stepwise variable elimination in the varus group.

**Table 7 jcm-09-04093-t007:** Results of univariable and multivariable linear regression analysis using correction rate (MAD) as a response in valgus and varus groups.

	Univariable Analysis				Multivariable Analysis		
Predictor (Response = Rate of Correction (MAD))	Kind of Deformity	Coef. (95% Cis)	St. Coef.	*p*−Value	Coef. (95% Cis)	St. Coef.	*p*−Value
Age at operation (years)	Valgus	−0.922 (−1.369 to −0.475)	−0.529	<0.001			
	Varus	−0.166 (−0.797 to 0.465)	−0.120	0.613			
Sex							
M:F	Valgus	0.353 (−1.233 to 1.940)	0.067	0.665			
	Varus	0.911 (−0.815 to 2.637)	0.237	0.315			
Weight (kg)	Valgus	−0.085 (−0.133 to −0.038)	−0.476	0.001			
	Varus	−0.057 (−0.161 to 0.048)	−0.242	0.304			
* Height (cm)	Valgus	−0.143 (−0.197 to −0.089)	−0.626	<0.001	−0.116 (−0.172 to −0.060)	−0.510	<0.001
	Varus	−0.023 (−0.074 to 0.028)	−0.201	0.395			
BMI	Valgus	−0.081 (−0.270 to 0.107)	−0.129	0.402			
	Varus	0.001 (−0.190 to 0.193)	0.003	0.990			
** Screw Location							
PT:DF	Valgus	−0.035 (−2.728 to 2.658)	−0.004	0.980			
	Varus	−2.122 (−3.905 to −0.338)	−0.482	0.032	−2.018 (−3.559 to −0.478)	−0.458	0.021
** Limb							
Rt:Lt	Valgus	−0.539 (−2.080 to 1.002)	−0.105	0.497			
	Varus	1.077 (−0.473 to 2.627)	0.306	0.190	0.996 (−0.174 to 2.167)	0.283	0.116
* Tip ratio AP	Valgus	−6.951 (−15.341 to 1.438)	−0.243	0.112	−9.375 (−16.121 to −2.630)	−0.328	0.100
	Varus	0.852 (−6.596 to 8.300)	0.053	0.825			
** Tip ratio Lat	Valgus	3.059 (−4.684 to 10.802)	0.119	0.443			
	Varus	−6.750 (−11.838 to −1.662)	−0.523	0.018	−4.238 (−8.828 to 0.352)	−0.328	0.090
* Size of Screw (mm)							
4.5:7.0	Valgus	1.922 (0.480 to 3.364)	0.374	0.012	1.393 (0.060 to 2.725)	0.271	0.047
	Varus	−0.850 (−2.439 to 0.739)	−0.240	0.308			

Coef, coefficient of linear regression; St. Coef, standardized coefficient of linear regression; Cis, confidence intervals M, Male; F, Female; PT, proximal tibia; DF, distal femur; Rt, right; Lt, left * Predictors were used as input variables for the multivariable analysis after backward stepwise variable elimination in the valgus group. ** Predictors were used as input variables for the multivariable analysis after backward stepwise variable elimination in the varus group.

**Table 8 jcm-09-04093-t008:** Comparison of correction rates between the group screwed at the distal femur and proximal tibia, before and after adjustment by covariates.

Measurement of Correction Rate	Statistics	Distal Femur	Proximal Tibia	*p*-Value
aTFA (°/month)	Two Sample *t*-test	1.05 ± 0.07	0.59 ± 0.08	<0.001
	ANCOVA *	1.195 ± 0.055	0.600 ± 0.077	<0.001
MAD (%/month)	Two Sample *t*-test	4.34 ± 0.38	3.15 ± 0.44	0.066
	ANCOVA *	5.092 ± 0.309	2.893 ± 0.433	<0.001

aTFA, anatomical tibiofemoral Angle; MAD, mechanical axis deviation * ANCOVA: analysis of covariance where sex, limb side, height, BMI, Tip ratio AP, Tip ratio Lat, and size of screw are used as covariates to be adjusted.

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
