# Peer review of "Affecting Factors and Correction Ratio in Genu Valgum or Varum Treated with Percutaneous Epiphysiodesis Using Transphyseal Screws"

_jcm, 2020, doi:10.3390/jcm9124093_

Round 1
Reviewer 1 Report
This is a comprehensive, well documented study. As stated the rate of correction has 3 variables : age, height, weight. You did not mention gender. Because girls mature about 2 years earlier than boys, surely, this would factor into calculation of the ideal timing for guided growth. While the academic merits of this knowledge are commendable, it may not translate into a clinically useful tool in a busy practice.
References: Several of the references are rather old and they focus exclusively upon permanent epiphysiodesis, PETS, and stapling (no longer practiced in most countries). Conspicuously absent, though perhaps beyond the scope of this manuscript, is any discussion of the tension band plates (TBP) that have gained widespread popularity.
The figure showing the placement of screws in the coronal projection nicely demonstrates a limitation of PETS, namely that the fulcrum of correction is within the physis - as opposed to at the perimeter with said TBP. Conceptually, the latter offers a safer and more efficient location for maximal angular correction.Also evident is the undue prominence of the femoral screw under the VMO - something the patient no doubt would complain of.
The average age of intervention was 11.29 years. Was there a gender difference? The duration of implantation was surprisingly short (8 mos. average). You did not discuss rebound deformity and how that is managed.
Author Response
This is a comprehensive, well documented study. As stated the rate of correction has 3 variables : age, height, weight. You did not mention gender. Because girls mature about 2 years earlier than boys, surely, this would factor into calculation of the ideal timing for guided growth. While the academic merits of this knowledge are commendable, it may not translate into a clinically useful tool in a busy practice.
Response 1: Thank you for your appropriate and accurate point. The male and female ratios were 23 and 41, respectively, and the ratios in the valgus group and varus group are also summarized in Table 1. And after controlling sex and other factors with a controlling demigraphic factor to make a multiple regression model, the correction ratio of aTFA and MAD was calculated. (result section line 142-148)
References: Several of the references are rather old and they focus exclusively upon permanent epiphysiodesis, PETS, and stapling (no longer practiced in most countries). Conspicuously absent, though perhaps beyond the scope of this manuscript, is any discussion of the tension band plates (TBP) that have gained widespread popularity. The figure showing the placement of screws in the coronal projection nicely demonstrates a limitation of PETS, namely that the fulcrum of correction is within the physis - as opposed to at the perimeter with said TBP. Conceptually, the latter offers a safer and more efficient location for maximal angular correction. Also evident is the undue prominence of the femoral screw under the VMO - something the patient no doubt would complain of.
Response 2: Thanks for your advice. In this study, the goal was to find out the correction ratio and affection factor of PETS surgery, and all cases used the growth guide technique using PETS surgery. Although the effectiveness of the tension band plate surgery was proven, it was not cited because it may obscure the argument, not the technique used in this study.
The average age of intervention was 11.29 years. Was there a gender difference? The duration of implantation was surprisingly short (8 mos. average). You did not discuss rebound deformity and how that is managed.
Response 3: That is the correct point. There was a difference between men and women at the start of PETS surgery, and it was additionally described in the result section line 111-112. The duration of implantation was averaged 8 months, but the range varied from 4 to 18 months, and was added to the result section line 137 and table1. We will conduct additional research on postoperative complications and rebound deformity. Thank you.
Reviewer 2 Report
Good manuscript. Interesting and important to readers.
Further important measurements around the knee should be added before publication. This is very important. The mLDFA and the mMPTA are essential factors. Please measure these two angles and add a suitable table to the manuscript. Please analyze not only the long leg axis in the frontal plane, but also the knee joint angles including the two mentioned above. Is the long-leg axis correct and is the joint line also correct? Be aware: You can produce a normal long-leg axis but have pathologic angles at the distal femur/proximal tibia and joint line.
Author Response
Good manuscript. Interesting and important to readers.
Further important measurements around the knee should be added before publication. This is very important. The mLDFA and the mMPTA are essential factors. Please measure these two angles and add a suitable table to the manuscript. Please analyze not only the long leg axis in the frontal plane, but also the knee joint angles including the two mentioned above. Is the long-leg axis correct and is the joint line also correct? Be aware: You can produce a normal long-leg axis but have pathologic angles at the distal femur/proximal tibia and joint line
Response 1: Thank you for your appropriate and accurate point. We additionally measured aLDFA and aMPTA to measure the pathologic condition of the knee joint angle and added it to patients and method section lines 78-80. The amount of change before and after surgery was analyzed and added to table2. However, in this study, aLDFA and aMPTA were not used to calculate the correction ratio in order to clarify the goal of obtaining the total long axis that the angular deformity is corrected per month. Your point is correct, but we studied valgus deformity and varus deformity together. In valgus patients, there were only 4 cases of PETS on proximal tibia and on distal femur in varus patients. In this reason, there were limitations in analyzing the overall mechanical axis deviation using LDFA and MPTA. Thank you.
Round 2
Reviewer 2 Report
Good revision of the manuscript was done. The mLDFA and the mMPTA are standard when reporting about alignment of the lower extremities in the frontal plane. Therefore, I recommend measuring it additionally, if possible. Probably no full leg standing x-ray is available, therefore it was not measured. Is this correct?